# Cheap DNN Pruning with Performance Guarantees. / Conference Submissions

## Abstract

Recent DNN pruning algorithms have succeeded in reducing the number of parameters in fully connected layers, often with little or no drop in classification accuracy. However, most of the existing pruning schemes either have to be applied during training or require a costly retraining procedure after pruning to regain classification accuracy. In this paper we propose a cheap pruning algorithm based on difference of convex functions (DC) optimisation. We also provide theoretical analysis for the growth in the Generalization Error (GE) of the new pruned network. Our method can be used with any convex regulariser and allows for a controlled degradation in classification accuracy, while being orders of magnitude faster than competing approaches. Experiments on common feedforward neural networks show that for sparsity levels above $90\%$ our method achieves **10%** higher classification accuracy compared to Hard Thresholding.

## 1 Introduction

Recently, deep neural networks have achieved state-of-the art results in a number of machine learning tasks LeCun et al. (2015). Training such networks is computationally intensive and often requires dedicated and expensive hardware. Furthermore, the resulting networks often require a considerable amount of memory to be stored. Using a Pascal Titan X GPU the popular AlexNet and VGG-16 models require 13 hours and 7 days, respectively, to train, while requiring 200MB and 600MB, respectively, to store. The large memory requirements limit the use of DNNs in embedded systems and portable devices such as smartphones, which are now ubiquitous.

A number of approaches have been proposed to reduce the DNN size during training time, often with little or no degradation to classification performance. Approaches include introducing bayesian, sparsity-inducing priors Louizos et al. (2017) Blundell et al. (2015) Molchanov et al. (2017) and binarization Hou et al. (2016) Courbariaux et al. (2016).Other methods include the hashing trick used in Chen et al. (2015), tensorisation Novikov et al. (2015) and efficient matrix factorisations Yang et al. (2015).

However, trained DNN models are used by researchers and developers that do not have dedicated hardware to train them, often as general feature extractors for transfer learning. In such settings it is important to introduce a *cheap* compression method, i.e., one that can be implemented as a postprocessing step with little or no retraining. Some first work in this direction has been Kim et al. (2015) Han et al. (2015a) Han et al. (2015b) although these still require a lengthy retraining procedure. Closer to our approach recently in Aghasi et al. (2016) the authors propose a convexified layerwise pruning algorithm termed Net-Trim. Building upon Net-Trim, the authors in Dong et al. (2017) propose LOBS, an algorithm for layerwise pruning by loss function approximation.

Pruning a neural network layer introduces a pertubation to the latent signal representations generated by that layer. As the pertubated signal passes through layers of non-linear projections, the pertubation could become arbitrary large. In Aghasi et al. (2016) and Dong et al. (2017) the authors conduct a theoretical analysis using the Lipschitz properties of DNNs showing the stability of the latent representations, over the training set, after pruning. The methods employed have connections to recent work Sokolic et al. (2017) Bartlett et al. (2017) Neyshabur et al. (2017) that have used the

Lipschitz properties to analyze the Generalization Error (GE) of DNNs, a more useful performance measure.

## 1.1 CONTRIBUTIONS

In this work we introduce a cheap pruning algorithm for dense layers of DNNs. We also conduct a theoretical analysis of how pruning affects the Generalization Error of the trained classifier.

- We show that the sparsity-inducing objective proposed in Aghasi et al. (2016) can be cast as a difference of convex functions problem, that has an efficient solution. For a fully connected layer with input dimension $d_1$, output dimension $d_2$ and $N$ training samples, Net-Trim and LOBS scale like $\mathcal{O}(Nd_1^3)$ and $\mathcal{O}((N+d_2)d_1^2)$, respectively. Our iterative algorithm scales like $\mathcal{O}(K(N+\frac{Nk}{N+\sqrt{k}})\log(\frac{1}{\epsilon})d_1d_2)$, where $\epsilon$ is the precision of the solution, $k$ is related to the Lipschitz and strong convexity constants, $d_2 \ll d_1$ and $K$ is the outer iteration number. Emprirically, our algorithm is orders of magnitude faster than competing approaches. We also extend our formulation to allow retraining a layer with any convex regulariser.

- We build upon the work of Sokolic et al. (2017) to bound the GE of a DNN after pruning. Our theoretical analysis holds for any bounded pertubation to one or multiple hidden DNN layers and provides a principled way of pruning while managing the GE.

Experiments on common feedforward architectures show that our method is orders of magnitude faster than competing pruning methods, while allowing for a controlled degradation in GE.

## 1.2 NOTATION AND DEFINITIONS

We use the following notation in the sequel:matrices ,column vectors, scalars and sets are denoted by boldface upper-case letters ($\boldsymbol{X}$), boldface lower-case letters ($\boldsymbol{x}$), italic letters ($x$) and calligraphic upper-case letters ($\mathcal{X}$), respectively. The covering number of $\mathcal{X}$ with $d$-metric balls of radius $\rho$ is denoted by $\mathcal{N}(\mathcal{X}; d, \rho)$. A $C_M$-regular $k$-dimensional manifold, where $C_M$ is a constant that captures "intrinsic" properties, is one that has a covering number $\mathcal{N}(\mathcal{X}; d, \rho) = (\frac{C_M}{\rho})^k$.

## 2 OUR FORMULATION

### 2.1 DC DECOMPOSITION

We consider a classification problem, where we observe a vector $\boldsymbol{x} \in \mathcal{X} \subseteq \mathbb{R}^N$ that has a corresponding class label $y \in \mathcal{Y}$. The set $\mathcal{X}$ is called the input space, $\mathcal{Y} = \{1, 2, ..., N_\mathcal{Y}\}$ is called the label space and $N_\mathcal{Y}$ denotes the number of classes. The samples space is denoted by $\mathcal{S} = \mathcal{X} \times \mathcal{Y}$ and an element of $\mathcal{S}$ is denoted by $s = (\boldsymbol{x}, y)$. We assume that samples from $\mathcal{S}$ are drawn according to a probability distribution $P$ defined on $\mathcal{S}$. A training set of $m$ samples drawn from $P$ is denoted by $S_m = \{s_i\}_{i=1}^m = \{(\boldsymbol{x}_i, y_i)\}_{i=1}^m$.

We start from the Net-Trim formulation and show that it can be cast as a difference of convex functions problem. For each training signal $\boldsymbol{x} \in \mathbb{R}^N$ we assume also that we have access to the inputs $\boldsymbol{a} \in \mathbb{R}^{d_1}$ and the outputs $\boldsymbol{b} \in \mathbb{R}^{d_2}$ of the fully connected layer, with a rectifier non-linearity $\rho(x) = \mathbf{max}(0, x)$. The optimisation problem that we want to solve is then

$$\min_{\boldsymbol{U}} \frac{1}{m} \sum_{s_j \in \mathcal{S}_m} ||\rho(\boldsymbol{U}^T\boldsymbol{a}_j) - \boldsymbol{b}_j||_2^2 + \lambda\Omega(\boldsymbol{U}), \tag{1}$$

where $\lambda$ is the sparsity parameter. The term $||\rho(\boldsymbol{U}^T\boldsymbol{a}_j) - \boldsymbol{b}_j||_2^2$ ensures that the nonlinear projection remains the same for training signals. The term $\lambda\Omega(\boldsymbol{U})$ is the convex regulariser which imposes the desired structure on the weight matrix $\boldsymbol{U}$.

The objective in Equation 1 is non-convex. We show that the optimisation of this objective can be cast as a difference of convex functions (DC) problem. We assume just one training sample $\boldsymbol{x} \in \mathbb{R}^N$, for simplicity, with latent representations $\boldsymbol{a} \in \mathbb{R}^d$ and $\boldsymbol{b} \in \mathbb{R}^z$

$$
\begin{aligned}
||\rho(\boldsymbol{U}^T\boldsymbol{a}) - \boldsymbol{b}||_2^2 &+ \lambda\Omega(\boldsymbol{U}) \\
&= \sum_i [\rho(\boldsymbol{u_i}^T\boldsymbol{a}) - \boldsymbol{b_i}]^2 + \lambda\Omega(\boldsymbol{U}) \\
&= \sum_i [\rho^2(\boldsymbol{u_i}^T\boldsymbol{a}) - 2\rho(\boldsymbol{u_i}^T\boldsymbol{a})\boldsymbol{b_i} + \boldsymbol{b_i}^2] + \lambda\Omega(\boldsymbol{U}) \\
&= \sum_i [\rho^2(\boldsymbol{u_i}^T\boldsymbol{a}) + \boldsymbol{b_i}^2] + \lambda\Omega(\boldsymbol{U}) + \sum_i [-2\boldsymbol{b_i}\rho(\boldsymbol{u_i}^T\boldsymbol{a})] \\
&= \sum_i [\rho^2(\boldsymbol{u_i}^T\boldsymbol{a}) + \boldsymbol{b_i}^2] + \lambda\Omega(\boldsymbol{U}) + \sum_{\substack{i \\ b_i < 0}} [-2\boldsymbol{b_i}\rho(\boldsymbol{u_i}^T\boldsymbol{a})] + \sum_{\substack{i \\ b_i \geq 0}} [-2\boldsymbol{b_i}\rho(\boldsymbol{u_i}^T\boldsymbol{a})].
\end{aligned}
\tag{2}
$$

Notice that after the split the first term ($b_i < 0$) is convex while the second ($b_i \geq 0$) is concave. We note that $b_i \geq 0$ by definition of the ReLu and set

$$
g(\boldsymbol{U}; \boldsymbol{x}) = \sum_i [\rho^2(\boldsymbol{u_i}^T\boldsymbol{a}) + \boldsymbol{b_i}^2],
\tag{3}
$$

$$
h(\boldsymbol{U}; \boldsymbol{x}) = \sum_{\substack{i \\ b_i > 0}} [2\boldsymbol{b_i}\rho(\boldsymbol{u_i}^T\boldsymbol{a})].
\tag{4}
$$

Then by summing over all the samples we get

$$
\begin{aligned}
f(\boldsymbol{U}) &= \sum_j g(\boldsymbol{U}; \boldsymbol{x}_j) + \lambda\Omega(\boldsymbol{U}) - \sum_j h(\boldsymbol{U}; \boldsymbol{x}_j) \\
&= g(\boldsymbol{U}) + \lambda\Omega(\boldsymbol{U}) - h(\boldsymbol{U}),
\end{aligned}
\tag{5}
$$

which is difference of convex functions. The rectifier nonlinearity is non-smooth, but we can alleviate that by assuming a smooth approximation. A common choice for this task is $\rho(x) = \frac{1}{\beta}\log(1 + \exp(\beta x))$, with $\beta$ a positive constant.

## 2.2 Optimisation

It is well known that DC programs have efficient optimisation algorithms. We propose to use the DCA algorithm Tao & An (1997). DCA is an iterative algorithm that consists in solving, at each iteration, the convex optimisation problem obtained by linearizing $h(\cdot)$ (the non-convex part of $f = g - h$) around the current solution. Although DCA is only guaranteed to reach local minima the authors of Tao & An (1997) state that DCA often converges to the global minimum, and has been used succefully to optimise a fully connected DNN layer Fawzi et al. (2015). At iteration $k$ of DCA, the linearized optimisation problem is given by

$$
\arg\min_{\boldsymbol{U}}\{g(\boldsymbol{U}) + \lambda\Omega(\boldsymbol{U}) - Tr(\boldsymbol{U}^T\nabla h(\boldsymbol{U}^k))\},
\tag{6}
$$

where $\boldsymbol{U}^k$ is the solution estimate at iteration $k$. The detailed procedure is then given in algorithms 1 and 2. We assume that the regulariser is convex but possibly non-smooth in which case the optimisation can be performed using proximal methods.

---

**Algorithm 1** FeTa (Fast and Efficient Trimming Algorithm)

---

1: Choose initial point: $\boldsymbol{U}^0$
2: **for** k = 1,...,K **do**
3:      Compute $C \leftarrow \nabla h(\boldsymbol{U}^k)$.
4:      Solve with Algorithm 2 the convex optimisation problem:

$$\boldsymbol{U}^{k+1} \leftarrow \arg\min_{\boldsymbol{U}}\{g(\boldsymbol{U}) + \lambda\Omega(\boldsymbol{U}) - Tr(\boldsymbol{U}^T C)\} \tag{7}$$

5: **end for**
6: If $\boldsymbol{U}^{k+1} \approx \boldsymbol{U}^k$ return $\boldsymbol{U}^{k+1}$.

---

**Algorithm 2a** Prox-SGD

---

1: **Initialization**: $\boldsymbol{U} \leftarrow \boldsymbol{U}^k, \rho, t_0$
2: **for** t = 1,...,T **do**
3:      Choose $(\boldsymbol{A}, \boldsymbol{B})$ randomly chosen minibatch.
4:      $\rho_t \leftarrow \min(\rho, \rho\frac{t_0}{t})$
5:      $\boldsymbol{U} = \text{prox}_{\lambda\Omega(\cdot)}(\boldsymbol{U} - \rho_t[\nabla g_{\boldsymbol{A},\boldsymbol{B}}(\boldsymbol{U}) - \nabla Tr(\boldsymbol{U}^T C)])$
6: **end for**
7: Return $\boldsymbol{U}^{k+1} \leftarrow \boldsymbol{U}$

---

**Algorithm 2b** Acc-Prox-SVRG

---

1: **Initialization**: $\tilde{\boldsymbol{x}}_0 \leftarrow \boldsymbol{U}^k, \beta, \eta$
2: **for** s = 1,...,S **do**
3:      $\tilde{\boldsymbol{u}} = \nabla g(\tilde{\boldsymbol{x}}_s)$
4:      $\boldsymbol{x}_1 = \boldsymbol{y}_1 = \tilde{\boldsymbol{x}}_s$
5:      **for** t = 1,2,...,T **do**
6:          Choose $(\boldsymbol{A}, \boldsymbol{B})$ randomly chosen minibatch.
7:          $\boldsymbol{u}_t = \nabla g_{\boldsymbol{A},\boldsymbol{B}}(\boldsymbol{y}_t) - \nabla g_{\boldsymbol{A},\boldsymbol{B}}(\tilde{\boldsymbol{x}}_s) + \tilde{\boldsymbol{u}}$
8:          $\boldsymbol{x}_{t+1} = \text{prox}_{\eta h}(\boldsymbol{y}_t - \eta\boldsymbol{u}_t)$
9:          $\boldsymbol{y}_{t+1} = \boldsymbol{x}_{t+1} + \beta(\boldsymbol{x}_{t+1} - \boldsymbol{x}_t)$
10:     **end for**
11:      $\tilde{\boldsymbol{x}}_{s+1} = \boldsymbol{x}_{T+1}$
12: **end for**
13: Return $\boldsymbol{U}^{k+1} \leftarrow \tilde{\boldsymbol{x}}_{S+1}$

---

In order to solve the linearized problem we first propose to use Proximal Stochastic Gradient Descent (Prox-SG), which we detail in algorithm 2a. At each iteration a minibatch $\boldsymbol{A}$ and $\boldsymbol{B}$ is drawn. The gradient for the smooth part is calculated and the algorithm takes a step in that direction with step size $\rho_t$. At each iteration $\rho$ is updated as $\rho_t \leftarrow \min(\rho, \rho\frac{t_0}{t})$ where $t_0$ is a hyperparameter. Then the proximal operator for the non-smooth regulariser $\lambda\Omega(\cdot)$ is applied to the result. We find that for the outer iterations $K$ the values 5 to 15 are usually sufficient, while for the inner iterations $T = 150$ is usually sufficient.

Although Prox-SG is very efficient, it sometimes doesn't converge to a good solution. We therefore propose to use Accelerated Proximal SVRG (Acc-Prox-SVRG), which was presented in Nitanda (2014). We detail this method in Algorithm 2b. For most experiments we see a significant improvement over Prox-SG. The hyperparameters for Acc-Prox-SVRG are the acceleration parameter $\beta$ and the gradient step $\eta$. We have found that in our experiments, using $\beta = 0.95$ and $\eta \in \{0.001, 0.0001\}$ gives the best results.

We name our algorithm FeTa, Fast and Efficient Trimming Algorithm.

## 3 GENERALIZATION ERROR

### 3.1 GENERALIZATION ERROR OF PRUNED LAYER

Having optimized our pruned layer for the training set we want to see if it is stable for the test set. We denote $f^1(\cdot, \boldsymbol{W}^1)$ the original representation and $f^2(\cdot, \boldsymbol{W}^2)$ the pruned representation. We assume that after training $\forall s_i \in \mathcal{S}_m \, ||f^1(\boldsymbol{a_i}, \boldsymbol{W}^1) - f^2(\boldsymbol{a_i}, \boldsymbol{W}^2)||_2^2 \leq C_1$. Second, we assume that $\forall s \in \mathcal{S} \, \exists s_i \in \mathcal{S}_m \Rightarrow ||a - a_i||_2^2 \leq \epsilon$. Third, the linear operators in $\boldsymbol{W}^1$, $\boldsymbol{W}^2$ are frames with upper frame bounds $B_1$, $B_2$ respectively.

**Theorem 3.1.** *For any testing point $s \in \mathcal{S}$, the distance between the original representation $f^1(\boldsymbol{a}, \boldsymbol{W}^1)$ and the pruned representation $f^2(\boldsymbol{a}, \boldsymbol{W}^2)$ is bounded by $||f^1(\boldsymbol{a}, \boldsymbol{W}^1) - f^2(\boldsymbol{a}, \boldsymbol{W}^2)||_2^2 \leq C_2$ where $C_2 = C_1 + (B_1 + B_2)\epsilon$.*

the detailed proof can be found in Appendix A.

### 3.2 GENERALIZATION ERROR OF CLASSIFIER

In this section we use tools from the robustness framework Xu & Mannor (2012) to bound the generalization error of the new architecture induced by our pruning. We consider DNN classifiers defined as

$$g(\boldsymbol{x}) = \max_{i \in [N_y]} (f(\boldsymbol{x}))_i, \tag{8}$$

where $(f(\boldsymbol{x}))_i$ is the $i-$th element of $N_y$ dimensional output of a DNN $f : \mathbb{R}^N \to \mathbb{R}^{N_y}$. We assume that $f(\boldsymbol{x})$ is composed of $L$ layers

$$f(\boldsymbol{x}) = f_L(f_{L-1}(...f_1(\boldsymbol{x}, \boldsymbol{W}_1), ... \boldsymbol{W}_{L-1}), \boldsymbol{W}_L), \tag{9}$$

where $f_l(\cdot, \boldsymbol{W}_l)$ represents the $l-$th layer with parameters $\boldsymbol{W}_l$, $l = 1, ..., L$. The output of the $l-$th layer is denoted $\boldsymbol{z}^l$, i.e. $\boldsymbol{z}^l = f_l(\boldsymbol{z}^{l-1}, \boldsymbol{W}_l)$. The input layer corresponds to $\boldsymbol{z}^0 = \boldsymbol{x}$ and the output of the last layer is denoted by $\boldsymbol{z} = f(\boldsymbol{x})$. We then need the following two definitions of the classification margin and the score that we take from Sokolic et al. (2017). These will be useful later for measuring the generalization error.

**Definition 3.1.** *(Score). For a classifier $g(\boldsymbol{x})$ a training sample $s_i = (\boldsymbol{x}_i, y_i)$ has a score*

$$o(s_i) = o(\boldsymbol{x}_i, g(\boldsymbol{x}_i)) = \min_{j \neq g(\boldsymbol{x}_i)} \sqrt{2}(\delta_{g(\boldsymbol{x}_i)} - \delta_j)^T f(\boldsymbol{x}_i), \tag{10}$$

where $\delta_i \in \mathcal{R}^{N_y}$ is the Kronecker delta vector with $(\delta_i)_i = 1$, and $g(\boldsymbol{x}_i)$ is the output class for $s_i$ from classifier $g(\boldsymbol{x})$ which can also be $g(\boldsymbol{x}_i) \neq y_i$.

**Definition 3.2.** *(Training Sample Margin). For a classifier $g(\boldsymbol{x})$ a training sample $s_i = (\boldsymbol{x}_i, y_i)$ has a classification margin $\gamma(s_i)$ measured by the $l_2$ norm if*

$$g(\boldsymbol{x}) = g(\boldsymbol{x}_i); \quad \forall \boldsymbol{x} : ||\boldsymbol{x} - \boldsymbol{x}_i||_2 < \gamma(s_i). \tag{11}$$

The classification margin of a training sample $s_i$ is the radius of the largest metric ball (induced by the $l_2$ norm) in $\mathcal{X}$ centered at $\boldsymbol{x}_i$ that is contained in the decision region associated with the classification label $g(\boldsymbol{x}_i)$. Note that it is possible for a classifier to misclassify a training point $g(\boldsymbol{x}_i) \neq y_i$. We then restate a useful result from Sokolic et al. (2017).

**Corollary 3.1.1.** *Assume that $\mathcal{X}$ is a (subset of) $C_M$-regular $k$-dimensional manifold, where $\mathcal{N}(\mathcal{X}; d; \rho) \leq (\frac{C_M}{\rho})^k$. Assume also that the DNN classifier $g(\boldsymbol{x})$ achieves a lower bound to the classification score $o(\tilde{s}) < o(s_i)$, $\forall s_i \in S_m$ and take $l(g(\boldsymbol{x}_i), y_i)$ to be the $0 - 1$ loss. Then for any $\delta > 0$, with probability at least $1 - \delta$,*

$$GE(g) \leq A \cdot (\gamma)^{-\frac{k}{2}} + B, \tag{12}$$

where $A = \sqrt{\frac{\log{(2)} \cdot N_y \cdot 2^{k+1} \cdot (C_M)^k}{m}}$ and $B = \sqrt{\frac{2 \log{1/\delta}}{m}}$ can be considered constants related to the data manifold and the training sample size, and $\gamma = \frac{o(\tilde{s})}{\prod_i ||\boldsymbol{W}_i||_2}$.

We are now ready to state our main result.

**Theorem 3.2.** *Assume that $\mathcal{X}$ is a (subset of) $C_M$-regular k-dimensional manifold, where $\mathcal{N}(\mathcal{X}; d; \rho) \leq (\frac{C_M}{\rho})^k$. Assume also that the DNN classifier $g_1(\boldsymbol{x})$ achieves a lower bound to the classification score $o(\tilde{s}) < o(s_i)$, $\forall s_i \in S_m$ and take $l(g(\boldsymbol{x}_i), y_i)$ to be the $0-1$ loss. Furthermore assume that we prune classifier $g_1(\boldsymbol{x})$ on layer $i_\star$ using Algorithm 1, to obtain a new classifier $g_2(\boldsymbol{x})$. Then for any $\delta > 0$, with probability at least $1 - \delta$, when $(\gamma - \sqrt{C_2} \cdot \frac{\prod_{i>i_\star} ||\boldsymbol{W}_i||_2}{\prod_i ||\boldsymbol{W}_i||_2}) > 0$,*

$$GE(g_2) \leq A \cdot (\gamma - \sqrt{C_2} \cdot \frac{\prod_{i>i_\star} ||\boldsymbol{W}_i||_2}{\prod_i ||\boldsymbol{W}_i||_2})^{-\frac{k}{2}} + B, \tag{13}$$

where $A = \sqrt{\frac{\log{(2)} \cdot N_y \cdot 2^{k+1} \cdot (C_M)^k}{m}}$ and $B = \sqrt{\frac{2 \log{1/\delta}}{m}}$ can be considered constants related to the data manifold and the training sample size, and $\gamma = \frac{o(\tilde{s})}{\prod_i ||\boldsymbol{W}_i||_2}$.

The detailed proof can be found in Appendix B. The bound depends on two constants related to intrinsic properties of the data manifold, the regularity constant $C_M$ and the intrinsic data dimensionality $k$. In particular the bound depends exponentially on the intrinsic data dimensionality $k$. Thus more complex datasets are expected to lead to less robust DNNs. This has been recently observed empirically in Bartlett et al. (2017). The bound also depends on the spectral norm of the hidden layers $||\boldsymbol{W}_i||_2$. Small spectral norms lead to a larger base in $(\cdot)^{-\frac{k}{2}}$ and thus to tigher bounds.

With respect to pruning our result is quite pessimistic as the pruning error $\sqrt{C_2}$ is multiplied by the factor $\prod_{i>i_\star} ||\boldsymbol{W}_i||_2$. Thus in our analysis the $GE$ grows exponentially with respect to the remaining layer depth of the pertubated layer. This is in line with previous work Raghu et al. (2016) Han et al. (2015b) that demonstrates that layers closer to the input are much less robust compared to layers close to the output. Our algorithm is applied to the fully connected layers of a DNN, which are much closer to the output compared to convolutional layers.

We can extend the above bound to include pruning of multiple layers.

**Theorem 3.3.** *Assume that $\mathcal{X}$ is a (subset of) $C_M$-regular k-dimensional manifold, where $\mathcal{N}(\mathcal{X}; d; \rho) \leq (\frac{C_M}{\rho})^k$. Assume also that the DNN classifier $g_1(\boldsymbol{x})$ achieves a lower bound to the classification score $o(\tilde{s}) < o(s_i)$, $\forall s_i \in S_m$ and take $l(g(\boldsymbol{x}_i), y_i)$ to be the $0-1$ loss. Furthermore assume that we prune classifier $g_1(\boldsymbol{x})$ on all layers using Algorithm 1, to obtain a new classifier $g_2(\boldsymbol{x})$. Then for any $\delta > 0$, with probability at least $1 - \delta$, when $(\gamma - \frac{\sum_{i=0}^{L} \sqrt{C_{i2}} \prod_{j=i+1}^{L} ||\boldsymbol{W}_j||_2}{\prod_i ||\boldsymbol{W}_i||_2}) > 0$,*

$$GE(g_2) \leq A \cdot (\gamma - \frac{\sum_{i=0}^{L} \sqrt{C_{i2}} \prod_{j=i+1}^{L} ||\boldsymbol{W}_j||_2}{\prod_i ||\boldsymbol{W}_i||_2})^{-\frac{k}{2}} + B, \tag{14}$$

where $A = \sqrt{\frac{\log{(2)} \cdot N_y \cdot 2^{k+1} \cdot (C_M)^k}{m}}$ and $B = \sqrt{\frac{2 \log{1/\delta}}{m}}$ can be considered constants related to the data manifold and the training sample size, and $\gamma = \frac{o(\tilde{s})}{\prod_i ||\boldsymbol{W}_i||_2}$.

The detailed proof can be found in Appendix C. The bound predicts that when pruning multiple layers the GE will be much greater than the sum of the GEs for each individual pruning. We note also the generality of our result; even though we have assumed a specific form of pruning, the GE bound holds for any type of bounded pertubation to a hidden layer.

## 4 EXPERIMENTS

We make a number of experiments to compare FeTa with LOBS and NetTrim-ADMM. We test two versions of the algorithm, FeTa$_1$ optimised with Prox-SGD and FeTa$_2$ optimized with Acc-Prox-SVRG. All experiments were run on a MacBook Pro with CPU 2.8GHz Intel Core i7 and RAM 16GB 1600 MHz DDR3.

### 4.1 TIME COMPLEXITY

First we compare the execution time of FeTa with that of LOBS and NetTrim-ADMM. We set $\Omega(\boldsymbol{U}) = ||\boldsymbol{U}||_1$ and aim for 95% sparsity. We set $d_1$ to be the input dimensions, $d_2$ to be the output dimensions and $N$ to be the number of training samples. Assuming that each $g(\boldsymbol{U}; \boldsymbol{x}_j)$ is $L$-Lipschitz smooth and $g(\boldsymbol{U})$ is $\mu$-strongly convex, if we optimise for an $\epsilon$ optimal solution and set $k = L/\mu$, FeTa$_1$ scales like $\mathcal{O}(K\frac{1}{\mu\epsilon}d_1d_2)$ and FeTa$_2$ scales like $\mathcal{O}(K(N + \frac{Nk}{N+\sqrt{k}})\log(\frac{1}{\epsilon})d_1d_2)$. We obtain this by multiplying the number of outer iterations $K$ with the number of gradient evaluations required to reach an $\epsilon$ good solution in inner Algorithm 2a and inner Algorithm 2b, and finally multiplying with the gradient evaluation cost. Conversely LOBS scales like $\mathcal{O}((N + d_2)d_1^2)$ while NetTrim-ADMM scales like $\mathcal{O}(Nd_1^3)$ due to the required Cholesky factorisation. This gives a computational advantage to our algorithm in settings where the input dimension is large. We validate this by constructing a synthetic experiment with $d_2 = 10$, $d_1 = \{2000 : 100 : 3000\}$ and $N = 1000$. The samples $\boldsymbol{a} \in \mathbb{R}^{d1}$ and $\boldsymbol{b} \in \mathbb{R}^{d2}$ are generated with $i.i.d$ Gaussian entries. We plot in Figure 1 the results, which are in line with the theoretical predictions.

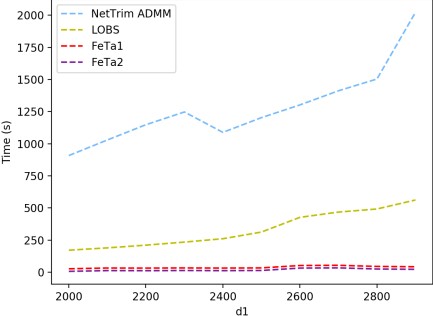

Figure 1: Time Complexity

### 4.2 CLASSIFICATION ACCURACY

#### 4.2.1 SPARSE REGULARISATION

In this section we perform experiments on the proposed compression scheme with feedforward neural networks. We compare the original full-precision network (without compression) with the following compressed networks: (i) FeTa$_1$ with $\Omega(\boldsymbol{U}) = ||\boldsymbol{U}||_1$ (ii) FeTa$_2$ with $\Omega(\boldsymbol{U}) = ||\boldsymbol{U}||_1$ (iii) Net-Trim (vi) LOBS (v) Hard Thresholding. We refer to the respective papers for Net-Trim and LOBS. Hard Thresholding is defined as $F(\boldsymbol{x}) = \boldsymbol{x} \odot I(|\boldsymbol{x}| > t)$, where $I$ is the elementwise indicator function, $\odot$ is the Hadamard product and $t$ is a positive constant.

Experiments were performed on two commonly used datasets:

1. *MNIST*: This contains $28 \times 28$ gray images from ten digit classes. We use 55000 images for training, another 5000 for validation, and the remaining 10000 for testing. We use the LeNet-5 model:

$$(1 \times 6C5) - MP2 - (6 \times 16C5) - MP2 - 120FC - 84FC - 10SM, \qquad (15)$$

    where $C5$ is a $5 \times 5$ ReLU convolution layer, $MP2$ is a $2 \times 2$ max-pooling layer, $FC$ is a fully connected layer and $SM$ is a linear softmax layer.

2. *CIFAR-10*: This contains 60000 $32 \times 32$ color images for ten object classes. We use 50000 images for training and the remaining 10000 for testing. The training data is augmented by random cropping to $24 \times 24$ pixels, random flips from left to right, contrast and brightness distortions to 200000 images. We use a smaller variant of the AlexNet model:

$$(3 \times 64C5) - MP2 - (64 \times 64C5) - MP2 - 384FC - 192FC - 10SM. \qquad (16)$$

We first prune **only the first** fully connected layer (the one furthest from the output) for clarity. Figure 2 shows the classification accuracy vs compression ratio for FeTa$_1$, FeTa$_2$, NetTrim, LOBS and Hard Thresholding. We see that Hard Thresholding works adequately up to $85\%$ sparsity. From this level of sparsity and above the performance of Hard Thresholding degrades rapidly and FeTa has $10\%$ higher accuracy on average. We also see a notable improvement of $3\% - 5\%$ for FeTa$_2$ over FeTa$_1$. Finally NetTrim and LOBS also give good results for a wide range of sparsity values, with LOBS giving the best results overall.

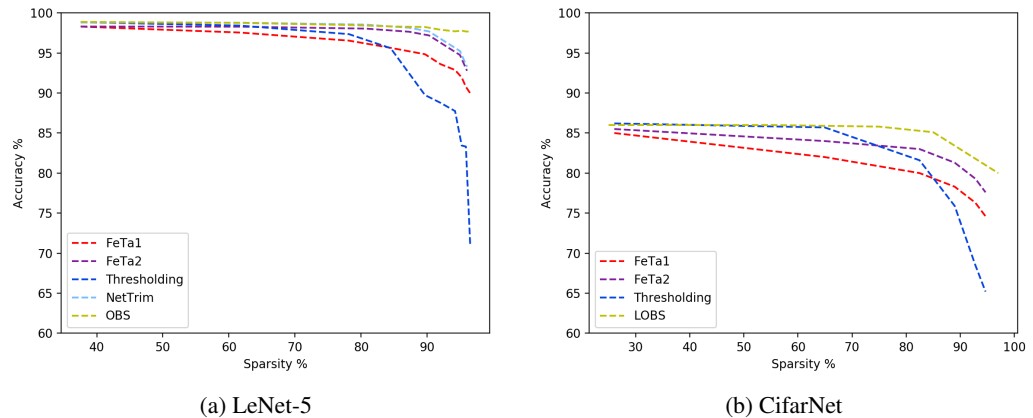

(a) LeNet-5            (b) CifarNet

Figure 2: Accuracy vs Sparsity

Table 1: Test accuracy rates (%) prune only first fully connected layer.

| Method | Networks | Original Accuracy | CR | Pruned Accuracy | Computation Time |
|---|---|---|---|---|---|
| Net-Trim | LeNet-5 | 99.2% | 95% | 95% | 455s |
| LOBS | LeNet-5 | 99.2% | 95% | 97% | 90s |
| Thresholding | LeNet-5 | 99.2% | 95% | 83% | - |
| FeTa$_1$ | LeNet-5 | 99.2% | 95% | **91%** | **32s** |
| FeTa$_2$ | LeNet-5 | 99.2% | 95% | **95%** | **18s** |
| Net-Trim | CifarNet | 86% | - | - | - |
| LOBS | CifarNet | 86% | 90% | 83.4% | 2h 47min |
| Thresholding | CifarNet | 86% | 90% | 73% | - |
| FeTa$_1$ | CifarNet | 86% | 90% | **77%** | **9min** |
| FeTa$_2$ | CifarNet | 86% | 90% | **80%** | **20min** |

Table 2: Test accuracy rates (%) prune all fully connected layers.

| Method | Networks | Original Accuracy | CR | Pruned Accuracy | Computation Time |
|---|---|---|---|---|---|
| Net-Trim | LeNet-5 | 99.2% | 90% | 95% | 500s |
| LOBS | LeNet-5 | 99.2% | 90% | 97% | 97s |
| Thresholding | LeNet-5 | 99.2% | 90% | 64% | - |
| FeTa$_1$ | LeNet-5 | 99.2% | 90% | **89%** | **41s** |
| FeTa$_2$ | LeNet-5 | 99.2% | 90% | **95%** | **38s** |
| Net-Trim | CifarNet | 86% | - | - | - |
| LOBS | CifarNet | 86% | 90% | 83.4% | 3h 15min |
| Thresholding | CifarNet | 86% | 90% | 64% | - |
| FeTa$_1$ | CifarNet | 86% | 90% | **68%** | **14min** |
| FeTa$_2$ | CifarNet | 86% | 90% | **71%** | **25min** |

For the task of pruning the first fully connected layer we also show detailed comparison results for all methods in Table 1. For the LeNet-5 model, FeTa achieves the same accuracy as Net-Trim while

being significantly faster. This is expected as the two algorithms optimise a similar objective, while FeTa exploits the structure of the objective to achieve lower complexity in optimisation. Furthermore FeTa achieves marginally lower classification accuracy compared to LOBS, and is significantly better than Thresholding. Overall FeTa enjoys competitive accuracy results while being able to prune the dense layer $5\times$ to $25\times$ faster compared to other approaches.

For the CifarNet model Net-Trim is not feasible on the machine used for the experiments as it requires over 16GB of RAM. Compared to LOBS FeTa again achieves marginally lower accuracy but is $8\times$ to $14\times$ faster. Note that as mentioned in Dong et al. (2017) and Wolfe et al. (2017) retraining can recover classification accuracy that was lost during pruning. Starting from a good pruning which doesn't allow for much degradation significantly reduces retraining time.

Next we prune both the fully connected layers in the two architectures to the same sparsity level and plot the results in Table 2. We lower the achieved sparsity for all methods to $90\%$. The accuracy results are mostly the same as when pruning a single layer, with FeTa achieving the same or marginally worse results while enjoying significant computation speedups for MNIST. At the same time there is a larger degradation for the CIFAR experiment. One significant difference is that Thresholding achieves a notably bad result of **64%** accuracy, which makes the method essentially inapplicable for multilayer pruning.

### 4.2.2 LOW RANK REGULARISATION

As a proof of concept for the generality of our approach we apply our method while imposing low-rank regularisation on the learned matrix $U$. For low rank $k$ we compare two methods (i) FeTa$_1$ with $\Omega(U) = ||U||_\star$ and optimised with Acc-Prox-SVRG and (ii) Hard Thresholding of singular values using the truncated SVD defined as $U = N\Sigma V^\star$, $\Sigma = \text{diag}(\{\sigma_i\}_{1\leq i\leq k})$. We plot the results in Figure 3.

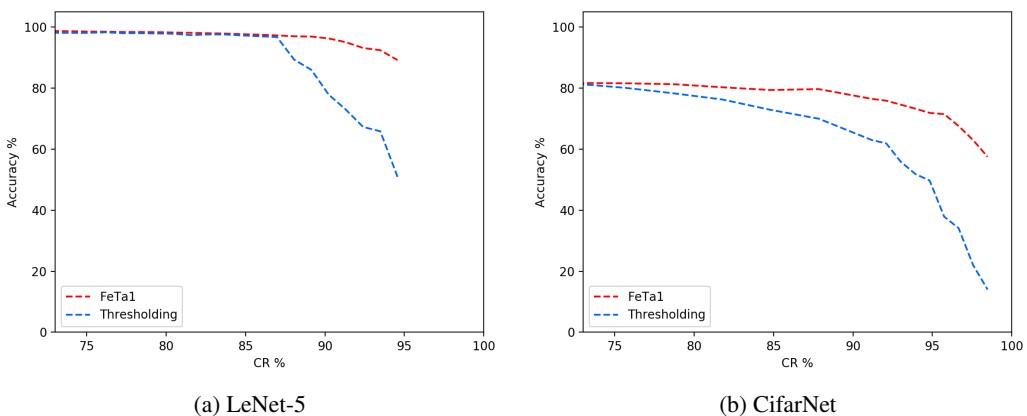

(a) LeNet-5          (b) CifarNet

Figure 3: Accuracy vs CR

In the above given $U \in \mathbb{R}^{d_1 \times d_2}$ the Commpression Ratio (CR) is defined as CR $= (k * d_1 + k + k * d_2)/(d_1 * d_2)$. The results are in line with the $l_1$ regularisation, with significant degredation in classification accuracy for Hard Thresholding above $85\%$ CR.

### 4.3 GENERALIZATION ERROR

According to our theoretical analysis the GE drops exponentially as the pruning moves away from the output layer. To corroborate this we train a LeNet-5 to high accuracy, then we pick a single layer and gradually increase its sparsity using Hard Thresholding. We find that the layers closer to the input are exponentially less robust to pruning, in line with our theoretical analysis. We plot the results in Figure 4.a. For some layers there is a sudden increase in accuracy around $90\%$ sparsity which could be due to the small size of the DNN. We point out that in empirical results Raghu et al. (2016) Han et al. (2015b) for much larger networks the degradation is entirely smooth.

Next we test our multilayer pruning bound. We prune to the same sparsity levels all layers in the sets $i \geq 0$, $i \geq 1$, $i \geq 2$, $i \geq 3$. We plot the results in Figure 4.b. It is evident that the accuracy loss for layer groups is not simply the addition of the accuracy losses of the individual layers, but shows an exponential drop in accordance with our theoretical result.

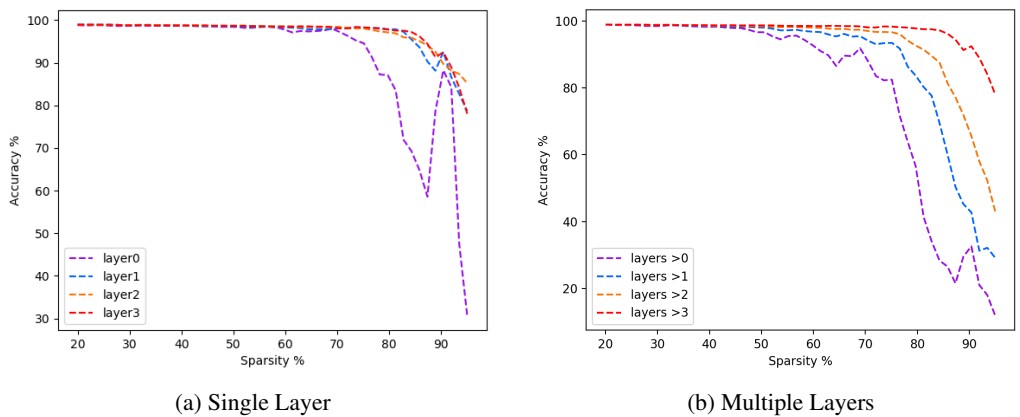

(a) Single Layer               (b) Multiple Layers

Figure 4: Layer Robustness

We now aim to see how well our bound captures this exponential behaviour. We take two networks $g_a$ pruned at layer 3 and $g_b$ pruned at layers 2 and 3 and make a number of simplifying assumptions. First we assume that in Theorem 3.3 $B = 0$ such that $GE(g_\star) \leq A \cdot (\gamma - \frac{\sum_{i=0}^{L} \sqrt{C_{i2}} \prod_{j=i+1}^{L} ||\boldsymbol{W}_j||_2}{\prod_i ||\boldsymbol{W}_i||_2})^{-\frac{k}{2}}$. This is logical as $B$ includes only log terms. Assuming that the bounds are tight we now aim to calculate

$$\frac{GE(g_a)}{GE(g_b)} = \left( \frac{\gamma - \sum_{i=0}^{L}(\sqrt{C_{i2}^a} / \prod_{j=0}^{i} ||\boldsymbol{W}_j||_2)}{\gamma - \sum_{i=0}^{L}(\sqrt{C_{i2}^b} / \prod_{j=0}^{i} ||\boldsymbol{W}_j||_2)} \right)^{-\frac{k}{2}}. \tag{17}$$

We know that the value of this ratio for 90% sparsity is $GE(g_a)/GE(g_b) \approx 0.1/0.4 = 1/4$ and we have managed to avoid the cumbersome $A$ parameter. Next we make the assumption that $k \approx 40$, this is common for the MNIST dataset and results from a simple dimensionality analysis using PCA. We also deviate slightly from our theory by using the average layerwise errors $\mathbb{E}_{s \sim S}[\sqrt{C_{i2}^a}]$, $\mathbb{E}_{s \sim S}[\sqrt{C_{i2}^b}]$, as well as the average scores $\mathbb{E}_{s \sim S}[o_a(\boldsymbol{x}, g_a(\boldsymbol{x}))]$, $\mathbb{E}_{s \sim S}[o_b(\boldsymbol{x}, g_b(\boldsymbol{x}))]$. We calculate

$$\frac{GE(g_a)}{GE(g_b)} \approx \frac{1}{5.6}, \tag{18}$$

which is very close to the empirical value.

## 5 CONCLUSION

In this paper we have presented an efficient pruning algorithm for fully connected layers of DNNs, based on difference of convex functions optimisation. Our algorithm is orders of magnitude faster than competing approaches while allowing for a controlled degradation in the Generalization Error. We provided a theoretical analysis of the degradation in GE resulting from our pruning algorithm. This analysis validates the previously observed phenomenon that network layers closer to the input are exponentially less robust to pruning compared to layers close to the output. Our theoretical analysis is of value by itself as it holds for any kind of bounded pertubation to one or multiple hidden DNN layers. Experiments on common feedforward architectures validate our results.

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

## 6 APPENDIX

A. PROOF OF THEOREM 3.1.

We denote by $f^1(\cdot, \boldsymbol{W}^1)$ the original representation and by $f^2(\cdot, \boldsymbol{W}^2)$ the pruned representation. We assume that, after training, $\forall s \in \mathcal{S}_m \ ||f^1(\boldsymbol{a}, \boldsymbol{W}^1) - f^2(\boldsymbol{a}, \boldsymbol{W}^2)||_2^2 \leq C_1$. Second, we assume that $\forall s \in \mathcal{S} \ \exists s_i \in \mathcal{S}_m \Rightarrow ||a - a_i||_2^2 \leq \epsilon$. Third the linear operators in $\boldsymbol{W}^1$, $\boldsymbol{W}^2$ are frames with upper frame bounds $B_1$, $B_2$ respectively. The following two lemmas will be useful:

**Lemma 6.1.** *The operator* $f^1(\cdot, \boldsymbol{W}^1)$ *is Lipschitz continuous with upper Lipschitz constant* $B_1$.

*Proof.* See Bruna et al. (2013) for details, the derivation is not entirely trivial due to the non-smoothness of the rectifier non-linearity. □

**Lemma 6.2.** *The operator* $f^2(\cdot, \boldsymbol{W}^2)$ *is Lipschitz continuous with upper Lipschitz constant* $B_2$.

*Proof.* We see that: $\frac{d}{dx}\rho(x) = \frac{d}{dx}\frac{1}{\beta}\log(1 + \exp(\beta x)) = \frac{\exp(\beta x)}{1+\exp(\beta x)} \leq 1$. Therefore the smooth approximation to the rectifier non-linarity is Lipschitz smooth with Lipschitz constant $k = 1$. Then $||f^2(x, \boldsymbol{W}^2) - f^2(y, \boldsymbol{W}^2)||_2^2 \leq k||\boldsymbol{W}^2 x - \boldsymbol{W}^2 y||_2^2 \leq ||\boldsymbol{W}^2(x - y)||_2^2 \leq B_2||x - y||_2^2$. □

We drop the $\boldsymbol{W}^i$ from the layer notation for clarity. Using the triangle inequality

$$
\begin{aligned}
||f^1(\boldsymbol{a}) - f^2(\boldsymbol{a})||_2^2 &= ||f^1(\boldsymbol{a}) + f^1(\boldsymbol{a}_i) - f^1(\boldsymbol{a}_i) - f^2(\boldsymbol{a})||_2^2 \\
&\leq ||f^1(\boldsymbol{a}) - f^1(\boldsymbol{a}_i)||_2^2 + ||f^1(\boldsymbol{a}_i) - f^2(\boldsymbol{a})||_2^2 \\
&= ||f^1(\boldsymbol{a}) - f^1(\boldsymbol{a}_i)||_2^2 + ||f^1(\boldsymbol{a}_i) + f^2(\boldsymbol{a}_i) - f^2(\boldsymbol{a}_i) - f^2(\boldsymbol{a})||_2^2 \\
&\leq ||f^1(\boldsymbol{a}) - f^1(\boldsymbol{a}_i)||_2^2 + ||f^1(\boldsymbol{a}_i) - f^2(\boldsymbol{a}_i)||_2^2 + ||f^2(\boldsymbol{a}_i) - f^2(\boldsymbol{a})||_2^2 \quad (19) \\
&\leq B_1||\boldsymbol{a}_i - \boldsymbol{a}||_2^2 + C + B_2||\boldsymbol{a}_i - \boldsymbol{a}||_2^2 \\
&= C_1 + (B_1 + B_2)||\boldsymbol{a}_i - \boldsymbol{a}||_2^2 \\
&\leq C_1 + (B_1 + B_2)\epsilon,
\end{aligned}
$$

where we used Lemma 6.1 and Lemma 6.2 in line 5.

B. PROOF OF THEOREM 3.2.

We will proceed as follows. We first introduce some prior results which hold for the general class of robust classifiers. We will then give specific prior generalization error results for the case of classifiers operating on datapoints from $C_m$-regular manifolds. Afterwards we will provide prior results for the specific case of DNN clasifiers. Finally we will prove our novel generalization error bound and provide a link with prior bounds.

We first formalize robustness for generic classifiers $g(\boldsymbol{x})$. In the following we assume a loss function $l(g(\boldsymbol{x}), y)$ that is positive and bounded $0 \leq l(g(\boldsymbol{x}), y) \leq M$.

**Definition 6.1.** *An algorithm* $g(\boldsymbol{x})$ *is* $(K, \epsilon(\mathcal{S}_m))$ *robust if* $\mathcal{S}$ *can be partitioned into $K$ disjoint sets, denoted by* $\{T_t\}_{t=1}^K$, *such that* $\forall s_i \in \mathcal{S}_m$, $\forall s \in \mathcal{S}$,

$$
s_i, s \in T_t, \Rightarrow |l(g(\boldsymbol{x}_i), y_i) - l(g(\boldsymbol{x}), y)| \leq \epsilon(\mathcal{S}_m). \quad (20)
$$

Now let $\hat{l}(\cdot)$ and $l_{\text{emp}}(\cdot)$ denote the expected error and the training error, i.e,

$$
\hat{l}(g) \triangleq \mathbb{E}_{s \sim S} l(g(\boldsymbol{x}), y); \quad l_{\text{emp}}(g) \triangleq \frac{1}{m} \sum_{s_i \in \mathcal{S}_m} l(q(\boldsymbol{x}_i), y_i) \quad (21)
$$

we can then state the following theorem from Xu & Mannor (2012):

**Theorem 6.3.** *If* $\mathcal{S}_m$ *consists of $m$ i.i.d. samples, and* $g(\boldsymbol{x})$ *is* $(K, \epsilon(\mathcal{S}_m))$-*robust, then for any* $\delta > 0$, *with probability at least* $1 - \delta$,

$$
GE(g) = |\hat{l}(g) - l_{emp}(g)| \leq \epsilon(\mathcal{S}_m) + M\sqrt{\frac{2K ln2 + 2ln(1/\delta)}{m}}. \quad (22)
$$

The above generic bound can be specified for the case of $C_m$-regular manifolds as in Sokolic et al. (2017). We recall the definition of the sample margin $\gamma(s_i)$ as well as the following theorem:

**Theorem 6.4.** *If there exists $\gamma$ such that*

$$\gamma(s_i) > \gamma > 0 \ \forall s_i \in S_m, \tag{23}$$

*then the classifier $g(\boldsymbol{x})$ is $(N_{\mathcal{Y}} \cdot \mathcal{N}(\mathcal{X}; d, \gamma/2), 0)$-robust.*

By direct substitution of the above result and the definiton of a $C_m$-regular manifold into Theorem 6.3 we get:

**Corollary 6.4.1.** *Assume that $\mathcal{X}$ is a (subset of) $C_M$ regular $k-$dimensional manifold, where $\mathcal{N}(\mathcal{X}; d, \rho) \leq (\frac{C_M}{\rho})^k$. Assume also that classifier $g(\boldsymbol{x})$ achieves a classification margin $\gamma$ and take $l(g(\boldsymbol{x}), y)$ to be the $0-1$ loss. Then for any $\delta > 0$, with probability at least $1 - \delta$,*

$$GE(g) \leq \sqrt{\frac{log(2) \cdot N_{\mathcal{Y}} \cdot 2^{k+1} \cdot (C_M)^k}{\gamma^k m}} + \sqrt{\frac{2log(1/\delta)}{m}}. \tag{24}$$

Note that in the above we have used the fact that $l(g(\boldsymbol{x}), y) \leq 1$ and therefore $M = 1$. The above holds for a wide range of algorithms that includes as an example SVMs. We are now ready to specify the above bound for the case of DNNs, adapted from Sokolic et al. (2017),

**Theorem 6.5.** *Assume that a DNN classifier $g(\boldsymbol{x})$, as defined in equation 8, and let $\tilde{\boldsymbol{x}}$ be the training sample with the smallest score $o(\tilde{s}) > 0$. Then the classification margin is bounded as*

$$\gamma(s_i) \geq \frac{o(\tilde{s})}{\prod_i \|\boldsymbol{W}_i\|_2} = \gamma. \tag{25}$$

We now prove our main result. We will denote by $\tilde{\boldsymbol{x}} = \arg \min_{s_i \in S_m} \min_{j \neq g(\boldsymbol{x}_i)} \boldsymbol{v}_{g(\boldsymbol{x}_i)j}^T f(\boldsymbol{x}_i)$ the training sample with the smallest score. For this training sample we will denote $j^{\star} = \arg \min_{j \neq g(\tilde{\boldsymbol{x}})} \boldsymbol{v}_{g(\tilde{\boldsymbol{x}})j}^T f(\tilde{\boldsymbol{x}})$ the second best guess of the classifier $g(\cdot)$. Throughout the proof, we will use the notation $\boldsymbol{v}_{ij} = \sqrt{2}(\boldsymbol{\delta}_i - \boldsymbol{\delta}_j)$.

First we assume the score $o_1(\tilde{\boldsymbol{x}}, g_1(\tilde{\boldsymbol{x}}))$ of the point $\tilde{\boldsymbol{x}}$ for the original classifier $g_1(\boldsymbol{x})$. Then, for the second classifier $g_2(\boldsymbol{x})$, we take a point $\boldsymbol{x}^{\star}$ that lies on the decision boundary between $g_2(\tilde{\boldsymbol{x}})$ and $j^{\star}$ such that $o_2(\boldsymbol{x}^{\star}, g_2(\tilde{\boldsymbol{x}})) = 0$. We assume for simplicity that, after pruning, the classification decisions do not change such that $g_1(\tilde{\boldsymbol{x}}) = g_2(\tilde{\boldsymbol{x}})$. We then make the following calculations

$$
\begin{aligned}
o_1(\tilde{\boldsymbol{x}}, g_1(\tilde{\boldsymbol{x}})) &= o_1(\tilde{\boldsymbol{x}}, g_1(\tilde{\boldsymbol{x}})) - o_2(\boldsymbol{x}^{\star}, g_2(\tilde{\boldsymbol{x}})) = \boldsymbol{v}_{g_1(\tilde{\boldsymbol{x}})j^{\star}}^T f^1(\tilde{\boldsymbol{x}}) - \boldsymbol{v}_{g_2(\tilde{\boldsymbol{x}})j^{\star}}^T f^2(\boldsymbol{x}^{\star}) \\
&= \boldsymbol{v}_{g_2(\tilde{\boldsymbol{x}})j^{\star}}^T (f^1(\tilde{\boldsymbol{x}}) - f^2(\boldsymbol{x}^{\star})) \\
&\leq \|\boldsymbol{v}_{g_2(\tilde{\boldsymbol{x}})j^{\star}}^T\|_2 \|f^1(\tilde{\boldsymbol{x}}) - f^2(\boldsymbol{x}^{\star})\|_2 = \|f_L^1(\tilde{\boldsymbol{x}}) - f_L^2(\boldsymbol{x}^{\star})\|_2 \\
&\leq \prod_{i>i^{\star}} \|\boldsymbol{W}_i\|_2 \|f_{i^{\star}}^1(\tilde{\boldsymbol{x}}) - f_{i^{\star}}^2(\boldsymbol{x}^{\star})\|_2 \\
&\leq \prod_{i>i^{\star}} \|\boldsymbol{W}_i\|_2 \{ \|f_{i^{\star}}^1(\tilde{\boldsymbol{x}}) - f_{i^{\star}}^1(\boldsymbol{x}^{\star})\|_2 + \|f_{i^{\star}}^1(\boldsymbol{x}^{\star}) - f_{i^{\star}}^2(\boldsymbol{x}^{\star})\|_2 \} \\
&\leq \prod_{i>i^{\star}} \|\boldsymbol{W}_i\|_2 \{ \|f_{i^{\star}}^1(\tilde{\boldsymbol{x}}) - f_{i^{\star}}^1(\boldsymbol{x}^{\star})\|_2 + \sqrt{C_2} \} \\
&\leq \prod_i \|\boldsymbol{W}_i\|_2 \|\tilde{\boldsymbol{x}} - \boldsymbol{x}^{\star}\|_2 + \prod_{i>i^{\star}} \|\boldsymbol{W}_i\|_2 \sqrt{C_2} \\
&\leq \prod_i \|\boldsymbol{W}_i\|_2 \gamma_2(s_i) + \prod_{i>i^{\star}} \|\boldsymbol{W}_i\|_2 \sqrt{C_2},
\end{aligned}
\tag{26}
$$

where we used Theorem 3.1 in line 5, since $\boldsymbol{x}^{\star}$ is not a training sample. From the above we can therefore write

$$\frac{o_1(\tilde{\boldsymbol{x}}, g_1(\tilde{\boldsymbol{x}})) - \sqrt{C_2} \prod_{i>i^{\star}} \|\boldsymbol{W}_i\|_2}{\prod_i \|\boldsymbol{W}_i\|_2} \leq \gamma_2(\tilde{\boldsymbol{x}}). \tag{27}$$

By following the derivation of the margin from the original paper Sokolic et al. (2017) and taking into account the definition of the margin we know that

$$\gamma = \frac{o_1(\tilde{\boldsymbol{x}}, g_1(\tilde{\boldsymbol{x}}))}{\prod_i \|\boldsymbol{W}_i\|_2} \leq \gamma_1(\tilde{\boldsymbol{x}}). \tag{28}$$

Therefore we can finally write

$$\gamma - \frac{\sqrt{C_2}\prod_{i>i^\star}||\boldsymbol{W}_i||_2}{\prod_i ||\boldsymbol{W}_i||_2} \leq \gamma_2(\tilde{\boldsymbol{x}}). \tag{29}$$

The theorem follows from direct application of Corollary 3.1.1. Note that if $\gamma - \frac{\sqrt{C_2}\prod_{i>i^\star}||\boldsymbol{W}_i||_2}{\prod_i ||\boldsymbol{W}_i||_2} < 0$ the derived bound becomes vacuous, as by definition $0 \leq \gamma_2(\tilde{\boldsymbol{x}})$.

## C. PROOF OF THEOREM 3.3.

We start as in theorem 3.2 by assuming the score $o_1(\tilde{\boldsymbol{x}}, g_1(\tilde{\boldsymbol{x}}))$ of the point $\tilde{\boldsymbol{x}}$ for the original classifier $g_1(\boldsymbol{x})$. Then, for the second classifier $g_2(\boldsymbol{x})$, we take a point $\boldsymbol{x}^\star$ that lies on the decision boundary between $g_2(\tilde{\boldsymbol{x}})$ and $j^\star$ such that $o_2(\boldsymbol{x}^\star, g_2(\tilde{\boldsymbol{x}})) = 0$. We assume as before that the classification decisions do not change such that $g_1(\tilde{\boldsymbol{x}}) = g_2(\tilde{\boldsymbol{x}})$. We write

$$
\begin{aligned}
o_1(\tilde{\boldsymbol{x}}, g_1(\tilde{\boldsymbol{x}})) &= o_1(\tilde{\boldsymbol{x}}, g_1(\tilde{\boldsymbol{x}})) - o_2(\boldsymbol{x}^\star, g_2(\tilde{\boldsymbol{x}})) = \boldsymbol{v}_{g_1(\tilde{\boldsymbol{x}})j^\star}^T f^1(\tilde{\boldsymbol{x}}) - \boldsymbol{v}_{g_2(\tilde{\boldsymbol{x}})j^\star}^T f^2(\boldsymbol{x}^\star) \\
&= \boldsymbol{v}_{g_2(\tilde{\boldsymbol{x}})j^\star}^T (f^1(\tilde{\boldsymbol{x}}) - f^2(\boldsymbol{x}^\star)) \\
&\leq ||\boldsymbol{v}_{g_2(\tilde{\boldsymbol{x}})j^\star}^T||_2 ||f^1(\tilde{\boldsymbol{x}}) - f^2(\boldsymbol{x}^\star)||_2 = ||f_L^1(\tilde{\boldsymbol{x}}) - f_L^2(\boldsymbol{x}^\star)||_2 \\
&\leq ||f_L^1(\tilde{\boldsymbol{x}}) - f_L^1(\boldsymbol{x}^\star)||_2 + ||f_L^1(\boldsymbol{x}^\star) - f_L^2(\boldsymbol{x}^\star)||_2 \\
&\leq ||f_L^1(\tilde{\boldsymbol{x}}) - f_L^1(\boldsymbol{x}^\star)||_2 + \sqrt{C_{L2}} \\
&\leq ||\boldsymbol{W}_L||_2 ||f_{L-1}^1(\tilde{\boldsymbol{x}}) - f_{L-1}^2(\boldsymbol{x}^\star)||_2 + \sqrt{C_{L2}} \\
&\leq ||\boldsymbol{W}_L||_2 \{||f_{L-1}^1(\tilde{\boldsymbol{x}}) - f_{L-1}^1(\boldsymbol{x}^\star)||_2 + ||f_{L-1}^1(\boldsymbol{x}^\star) - f_{L-1}^2(\boldsymbol{x}^\star)||_2\} + \sqrt{C_{L2}} \\
&\leq ||\boldsymbol{W}_L||_2 \{||f_{L-1}^1(\tilde{\boldsymbol{x}}) - f_{L-1}^1(\boldsymbol{x}^\star)||_2 + \sqrt{C_{L-1,2}}\} + \sqrt{C_{L2}} \\
&\leq ||\boldsymbol{W}_L||_2 ||f_{L-1}^1(\tilde{\boldsymbol{x}}) - f_{L-1}^1(\boldsymbol{x}^\star)||_2 + ||\boldsymbol{W}_L||_2\sqrt{C_{L-1,2}} + \sqrt{C_{L2}} \\
&\leq ... \\
&\leq \prod_i ||\boldsymbol{W}_i||_2 ||\tilde{\boldsymbol{x}} - \boldsymbol{x}^\star||_2 + \sum_{i=0}^L \sqrt{C_{i2}} \prod_{j=i+1}^L ||\boldsymbol{W}_j||_2 \\
&\leq \prod_i ||\boldsymbol{W}_i||_2 \gamma_2(s_i) + \sum_{i=0}^L \sqrt{C_{i2}} \prod_{j=i+1}^L ||\boldsymbol{W}_j||_2.
\end{aligned}
\tag{30}
$$

We can then write

$$\frac{o_1(\tilde{\boldsymbol{x}}, g_1(\tilde{\boldsymbol{x}})) - \sum_{i=0}^L \sqrt{C_{i2}}\prod_{j=i+1}^L ||\boldsymbol{W}_j||_2}{\prod_i ||\boldsymbol{W}_i||_2} \leq \gamma_2(\tilde{\boldsymbol{x}}). \tag{31}$$

Then as before

$$\gamma - \frac{\sum_{i=0}^L \sqrt{C_{i2}}\prod_{j=i+1}^L ||\boldsymbol{W}_j||_2}{\prod_i ||\boldsymbol{W}_i||_2} \leq \gamma_2(\tilde{\boldsymbol{x}}). \tag{32}$$

The theorem follows from direct application of Corollary 3.1.1.

