# OpenReview forum: "Cheap DNN Pruning with Performance Guarantees "
_ICLR.cc/2018/Conference — Reject_

### Official Review · AnonReviewer2 · 2017-11-26
**The manuscript mainly presents a cheap pruning algorithm for dense layers of DNNs. It reformulates the non-convex optimization problem in (Aghasi et al., 2016) as a difference of convex (DC) problem, which can be solved quite efficiently using the DCA algorithm (Tao and An, 1997). The contribution is valuable since the complexity is significantly reduced, but there are many syntax errors and the accuracy of the model is not satisfactory.**

**Rating:** 6
**Confidence:** 3

**Review:**

The manuscript mainly presents a cheap pruning algorithm for dense layers of DNNs. The proposed algorithm is an improvement of Net-Trim (Aghasi et al., 2016), which is to enforce the weights to be sparse.

The main contribution of this manuscript is that the non-convex optimization problem in (Aghasi et al., 2016) is reformulated as a difference of convex (DC) problem, which can be solved quite efficiently using the DCA algorithm (Tao and An, 1997). The complexity of the proposed algorithm is much lower than Net-Trim and its fast version LOBS (Dong et al., 2017). The authors also analyze the generalization error bound of DNN after pruning based on the work of (Sokolic et al., 2017).

Although this is an incremental work built upon (Aghasi et al., 2016) and an existing algorithm (Tao and An, 1997) is adopted for optimization, the contribution is valuable since the complexity is significantly reduced by utilizing the proposed difference of convex reformulation. Although the main idea is clearly presented, there are many syntax errors and I suggest the authors carefully checking the manuscript.

Pros:
1.	The motivation is clear and the presented reformulation is reasonable.

2.	The generalization error analysis and the conclusion of “layers closer to the input are exponentially less robust to pruning” is interesting.

Cons:
1.	There are many syntax errors, e.g., “Closer to our approach recently in Aghasi et al. (2016) the authors”, “an cheap pruning algorithm”, etc. Besides, there is no discussion for the results in Table 1.

2.	Although the complexity of the proposed method is much lower than the compared approaches (Net-Trim and LOBS), there seems to be a large sacrifice on accuracy. For example, the accuracy drops from 95.2% to 91% compared with Net-Trim in the LeNet-5 model and from 80.5% to 74.6% compared with LOBS in the CifarNet model. The proposed method is only better than hard-thresholding.

---

> ### Author Response · Authors · 2017-12-20
> **Reply to reviewer 2.**
>
> Thank you very much for your useful comments.
>
> We are currently working to improve the submitted work. We have improved the accuracy of the pruned architectures by using a different optimisation during the DCA iterations, which allows us to reach better minima. Specifically we have used Proximal Stochastic Variance Reduction (Prox-SVRG) instead of Proximal Stochastic Gradient Descent (Prox-SGD). We are also working to remove any syntax or spelling errors.

---

### Official Review · AnonReviewer1 · 2017-11-26
**Interesting use of DC functions**

**Rating:** 5
**Confidence:** 3

**Review:**

The problem of pruning DNNs is an active area of study.
This paper addresses this problem by posing the Net-trim objective function as  a Difference of convex(DC) function. This allows for an immediate application of DC function minimization using existing techniques. An analysis of Generalization error
is also given.

The main novelty seems to be the interesting connection to DC function minimization. The benefits seem to be a faster algorithm for pruning.

About the generalization error the term C_2 needs to be more well defined otherwise the coefficient of  A would be -ve which may lead to complications.

Experimental investigations are reasonable and the results are convincing.

A list of Pros:
1. Interesting connection to DC function
2. Attempt to analyze generalization error
3. Faster speed of convergence empirically

A list of Cons:
1. The contribution in posing the objective as a DC function looks limited as it is very straightforward. Also the algorithm is
direct application
2. The time complexity analysis is imprecise. Since the proposed algorithm is iterative time complexity would depend on the number of iterations.

---

> ### Author Response · Authors · 2018-01-05
> **Reply to reviewer 2.**
>
> Thank you very much for your useful comments.
>
> Our GE depends on a lower bound for the margin parameter \gamma. For a large enough C_2 we indeed have a lower bound on the margin \gamma> A where A<0. However the bound on the GE then becomes vacuous as by definition gamma > 0.  For consistency we have added the constraint that the base in the exponentiation needs to be positive, and thus the C_2 should be sufficiently small. Furthermore we have made a direct computation of the ratio between two GE bounds in section 4.3 which shows that the error C_2 is indeed small enough at least for the LeNet-5 architecture.
>
> Concerning the generality of our approach please note that our method can be applied with any convex regulariser, possibly with ones that do not aim at network compression, but for example for protection against adversarial examples. Also our theoretical analysis of the GE includes not only pruning but any type of bounded perturbation to one or multiple hidden layers.
>
> We have also added a more detailed analysis of the computational complexity of our algorithm. This includes the iterations "K" required by the outer DCA algorithm, as well as  the gradient evaluation number to reach an \epsilon good solution for the inner stochastic descent algorithm.

---

### Official Review · AnonReviewer3 · 2017-11-26
**This paper provides an interesting but incomplete analysis of a NetTrim-inspired pruning algorithm.**

**Rating:** 5
**Confidence:** 4

**Review:**

This paper casts the pruning optimization problem of NetTrim as a difference of convex problems, and uses DCA to obtain the smaller weight matrix; this algorithm is also analyzed theoretically to provide a bound on the generalization error of the pruned network.

However, there are many questions that aren't answered in the paper that make it difficult to evaluate: in particular, some experimental results leave open more questions for performance analysis.

Quality: of good quality, but incomplete.
Clarity: clear with some typos
Originality: a new approach to the NetTrim algorithm, which is somewhat original, and a new generalization bound for the algorithm.
Significance: somewhat significant.

PROS
- A very efficient algorithm for pruning, which can run orders of magnitude faster than the approaches that were compared to on certain architectures.
- An interesting generalization bound for the pruned network which is in line experimentally with decreasing robustness to pruning on layers close to the input.

CONS
- Non-trivial loss of accuracy on the pruned network, which cannot be estimated for larger-scale pruning as the experiments only prune one layer.
- No in-depth analysis of the generalization bound.

Main questions:
- You mention you use a variant of DCA: could you detail what differences Alg. 2 has with classical DCA?
- Where do you use the 0-1 loss in Thm. 3.2?
- I think your result in Theorem 3.2 would be significantly stronger if you could provide an analysis of the bound you obtain: in which cases can we expect certain terms to be larger or smaller, etc.
- Your experiments in section 4.2 show a non-trivial degradation of the accuracy with FeTa. Although the time savings seem worth the tradeoff to prune *one* layer, have you run the same experiments when pruning multiple layers? Could you comment on how the accuracy evolves with multiple pruned layers?
- It would be nice to see the curves for NetTrim and/or LOBS in Fig. 2.
- Have you tried retraining the network after pruning? Did you observe the same behavior as mentioned in (Dong et al., 2017) and (Wolfe et al., 2017)?
- It would be interseting to plot the theoretical (pessimistic) GE bound as well as the experimental accuracy degradation.

Nitpicks:
-Ubiquitous (first paragraph)
-difference of convex problemS
- The references should be placed before the appendix.
- The amount of white space should be reduced (e.g. around Eq. (1)).

---

> ### Author Response · Authors · 2017-12-20
> **Reply to reviewer 3.**
>
> Thank you very much for your useful comments.
>
> -The term "variant" is redundant and will be removed from the corrected version. It's usage was meant to convey that the minimisation of the linearised objective in the DCA iterations, is done using stochastic gradient descent.
>
> - The theoretical analysis of the generalisation error was based largely on Theorem 2 and Corollary 1 page 8 in [1] as well as Theorem 3 page 6 of [2]. It is in this last theorem that the assumption for the 0-1 loss is needed, the loss needs to be non-negative and upper bounded by a scalar M. We will restate all the relevant theorems in the Appendix for clarity, and provide some discussion on the proposed generalisation bound.
>
> -One cause for the non-trivial loss of accuracy is that we use Proximal Stochastic Gradient Descent for the optimisation. Prox-SGD fails to converge to a good solution within the given iterations. We propose to instead use Proximal Stochastic Variance Reduction (Prox-SVRG). This has so far improved our results. We are currently conducting more experiments to address this and other reviewer questions and to further validate our claims.
>
> -Please note that the GE bound depends on constants that are difficult to calculate for real data, such as the intrinsic data dimensionality "k". This in turn makes plotting the theoretical GE bound non-trivial. However the bound should give a good intuition about the behaviour of a DNN in relation to the margin "gamma" as the underlying assumptions that it makes have been tested empirically in [1] [3].
>
> [1] Sokolic, Jure, et al. "Robust large margin deep neural networks." IEEE Transactions on Signal Processing (2017).
> [2] Xu, Huan, and Shie Mannor. "Robustness and generalization." Machine learning 86.3 (2012): 391-423.
> [3] Sokolic, Jure, et al. "Generalization Error of Invariant Classifiers." arXiv preprint arXiv:1610.04574 (2016).

---

### Author Response · Authors · 2018-01-05
**Reply to all reviewers.**

Thank you very much for your useful comments.

We have made a number of changes to the original submission.
- We have applied a different optimisation scheme for the optimisation of the linearised objective, specifically Proximal SVRG with acceleration. This has improved the accuracy for the DNNs after pruning with the proposed algorithm.
- We have made multilayer pruning experiments on the architectures tested originally.
- We have generalised our theoretical analysis to pruning multiple hidden layers, and have tested the validity of our analysis through the direct computation of a ratio between two GEs.
-We have also addressed other reviewer comments by providing additional analysis of our theoretical and experimental results, and fixing other minor issues.

We apologise for any whitespace or syntax errors, which have been corrected to the best of our ability, and kindly ask that the reviewers reconsider their decision.

---

### Decision · Program_Chairs · 2018-01-29
**ICLR 2018 Conference Acceptance Decision**

**Decision:**

Reject

**Comment:**

Dear authors,

While the reviewers appreciated the idea, the significant loss of accuracy was a concern. Even though you made significant changes to the submission, it is unfortunately unrealistic to ask the reviewers to do another review of a heavily modified version in such a short amount of time.

Thus, I cannot accept this paper for publication but I encourage you to address the reviewers' concerns and resubmit at a later conference.